# The Current Epizootiological Situation of Three Major Viral Infections Affecting Cattle in Egypt

**DOI:** 10.3390/v16101536

**Published:** 2024-09-28

**Authors:** Sherin R. Rouby, Ahmed H. Ghonaim, Xingxiang Chen, Wentao Li

**Affiliations:** 1Department of Veterinary Medicine, Faculty of Veterinary Medicine, Beni-Suef University, Beni-Suef 62511, Egypt; shereen.rouby@vet.bsu.edu.eg; 2National Key Laboratory of Agricultural Microbiology, College of Veterinary Medicine, Huazhong Agricultural University, Wuhan 430070, China; drcahmed91@gmail.com; 3Department of Animal and Poultry Health, Desert Research Center, Cairo 11435, Egypt; 4College of Veterinary Medicine, Nanjing Agricultural University, Nanjing 210095, China; cxx@njau.edu.cn; 5The Cooperative Innovation Center for Sustainable Pig Production, Wuhan 430070, China

**Keywords:** cattle, bovine ephemeral fever, foot and mouth disease, lumpy skin disease

## Abstract

One of the major factors hindering efficient livestock production is the presence of high-impact infectious animal diseases, such as foot and mouth disease (FMD), lumpy skin disease (LSD), and bovine ephemeral fever (BEF), which are notable viral infections affecting cattle in Egypt, leading to significant economic losses. FMD is caused by the foot and mouth disease virus (FMDV) of the genus *Aphthovirus* in the *Picornaviridae* family. LSD is caused by lumpy skin disease virus (LSDV) of *Capripox* genus within the *Poxviridae* family, subfamily *Chordopoxvirinae*. BEF is caused by bovine ephemeral fever virus (BEFV) of genus *Ephemerovirus* in the *Rhabdoviridae* family. FMD is a highly contagious viral infection of domestic and wild cloven-hooved animals and can spread through the wind. On the other hand, LSD and BEF are arthropod-borne viral diseases that mainly affect domestic cattle and water buffalo. Despite government vaccination efforts, these three viral diseases have become widespread in Egypt, with several reported epidemics. Egypt’s importation of large numbers of animals from different countries, combined with unregulated animal movements through trading and borders between African countries and Egypt, facilitates the introduction of new FMDV serotypes and lineages not covered by the current vaccination plans. To establish an effective control program, countries need to assess the real epizootic situation of various infectious animal diseases to develop an efficient early warning system. This review provides information about FMD, LSD, and BEF, including their economic impacts, causative viruses, global burden, the situation in Egypt, and the challenges in controlling these diseases.

## 1. Introduction

Livestock is a crucial part of Egyptian agriculture, with cattle and buffaloes being the primary sources of milk and red meat in the country. As living standards rise and awareness of the importance of animal proteins grows, there is an increased demand for livestock products. However, efficient livestock production is hindered by the prevalence of highly impactful infectious animal diseases. These diseases lead to significant economic losses through decreased milk and meat production, as well as issues like infertility, abortion, deaths, and treatment costs [1]. Some of the most threatening viruses to animal health include foot and mouth disease (FMD), lumpy skin disease (LSD), and bovine ephemeral fever (BEF). These diseases are considered endemic in Egypt. Foot and mouth disease virus (FMDV) exists in seven divergent serotypes, while only one serotype of lumpy skin disease virus (LSDV) and bovine ephemeral fever virus (BEFV) has been identified so far. This article aims to discuss these three viral diseases in terms of their economic importance, causative virus, global burden, and the epizootic situation of each in Egypt, including reporting on outbreaks and circulating strains.

## 2. Foot and Mouth Disease

### 2.1. General Information

FMD is one of the most important contagious viral diseases affecting cloven-hoofed animals (order: Artiodactyla), including cattle, buffaloes, pigs, sheep, and goats [2]. FMD is the paradigm of a transboundary animal disease and is considered the most serious challenge for livestock’s health not only in Egypt but also in about two-thirds of the countries in the world where the disease is endemic [3]. FMD tends to be more severe in cattle and pigs, while sheep and goats may sometimes exhibit milder symptoms that can be undiagnosed. Notable clinical signs include fever and the presence of vesicles (fluid-filled blisters) on the feet, particularly between the toes and on the heels, as well as around the mouth, especially on the lips, tongue, palate, and mammary glands [4]. However, the most characteristic lesions are typically observed in the interdigital spaces (between the hooves) and coronary bands of the hooves [5] (Figure 1).

### 2.2. Economic Impact

FMD causes severe economic losses at the farmer level in terms of reducing the productivity of adult animals (red meat, milk, and milk byproducts) and increasing the mortality of young animals due to myocarditis. At the national level, FMD affects the economy due to the extreme costs of control measures in endemic countries (vaccination campaigns, surveillance programs, restrictions on animal movement, and the closure of animal markets) [6] (Figure 2).

### 2.3. The Virus

Foot and mouth disease virus (FMDV), a single-stranded, positive-sense RNA, belongs to the genus *Aphthovirus* of the family *Picornaviridae*, order *Picornavirales* [7]. The FMD genome contains a single open reading frame (ORF) encoding a large polyprotein, which, after translation, is separated into four regions on the basis of mature protein functions (L, P1, P2, and P3). The P1 region codes for four structural proteins involved in capsid formation, whereas the L, P2, and P3 regions code for eight nonstructural proteins involved in virus genome replication and protein cleavage activities [8].

Genetic variation is highly important when it occurs in the capsid-encoding area because it may result in the development of new antigenic variants [9].

The virus exists in seven serologically and genetically divergent serotypes: O, A, C, southern African territories (SAT1–3), and Asia-1 [10]. Within each serotype, there are numerous topotypes that are additionally divided into several genotypes (lineages or strains) that are attributed to a high rate of mutation (10^−3^ to 10^−5^ per nucleotide site per replication) from error-prone RNA replication, recombination, and host selection [11]. There is no cross-protection between diverse serotypes of FMDV; however, there is a variable degree of cross-protection among the subtypes within the same serotype [12].

### 2.4. Global Burden

The global clustering of FMD viruses has been divided into seven virus pools (three pools cover Europe, the Middle East and Asia; three pools cover Africa; and one pool covers the Americas), where numerous serotypes occur but within which there are topotypes that remain mostly limited to that pool [13]. The most prevalent serotypes are FMDV serotypes A and O, which have been reported in Asia, Africa, and South America, whereas serotypes SAT1, 2, and 3 currently circulate in sub-Saharan Africa, North Africa, and the Middle East. Serotype Asia1 exists on the Asian continent [13]. Serotype C has not triggered any outbreaks since it was documented in 2004 in Kenya and Brazil [13].

### 2.5. Egyptian Situation

Foot and mouth disease has been reported to be endemic in Egypt since 1950 according to Knowles et al. [14] and Pattnaik et al. [15] (Table 1). One of the FMDV serotypes that has a long history in Egypt is serotype O. It was the only serotype described in Egypt between 1964 and 2005, with the exception of an outbreak in 1972 linked to FMDV serotype A [14]. FMDV serotype A (A/Africa/G-VII) was introduced in 2006 through animal import from East Africa and has caused severe economic losses; around 27 outbreaks have ensued in 24 Egyptian governorates, and approximately 11,780 animals have suffered [14]. According to the OIE/FAOWRLFMD Annual Report (2012), FMDV-A/Asia/iran05, FMDV-O/ME-SA/PanAsia-2, and FMDV-O/EA-3 were isolated from 2010 and 2011. In 2012, FMDV serotype SAT2 was recorded as an exotic viral strain associated with a dramatic upsurge of FMD outbreaks across the country (about 40 outbreaks of FMDV/SAT2 resulted in more than 60,000 clinical cases and more than 14,000 mortalities in calves) and appeared in two lineages, namely Alx-12 and Ghb-12, which both belong to topotype VII (SAT2/VII/Ghb-12 and SAT2/VII/Alx-12) [16,17]. Since 2013, three FMDV serotypes (O, A, and SAT2) have been circulating in different localities in Egypt [18,19]. In 2016, FMDV-O/ME-SA/Sharqia-72 was identified along with FMDV-O/EA-3 and FMDV-A/Africa/G IV [18].

At the end of 2018, severe stomatitis and mortality among cattle and buffaloes were reported in three governorates (Alexandria, Ismailia, and Sharquia) [20]. Through sequencing and phylogenetic analysis, the circulation of FMDV-SAT2 (topotype VII, Lib12 lineage) was confirmed in all the collected samples [20].

Eleven various FMDV lineages have been reported in Egypt since 2006; within serotype O: O/ME-SA/Sharquia-72 and PanAsia-2 as well as O/EA-3/Qal-13, Ism-16, and Alx-17; within serotype A: A/Africa/G-VII KEN−05, A/Africa/G-IV ISM-12, A/Asia/Iran-05BAR−08, and the historic A/Africa/G-II; and for serotype SAT2: SAT2/VII/Ghb-12 and Alx-12 as well as the recently identified SAT2/VII/Lib-12 (in Egypt in 2018) [20] (Table 1, Figure 3).

In 2020, FMDV/A/G-IV appeared to be the predominant serotype circulating in the country and was found to have a genetic relationship with the ancestor Sudan type. In Jan and Feb 2021, four confirmed outbreaks of FMDV-A/AFRICA/G-IV were reported in Dakhalia Governorate (2), Sohag Governorate (1), and Kafrelshikh Governorate (1) [21]. In 2022, FMDV serotype O, the Euro-SA topotype in Egypt, was detected in clinical samples collected from one farm [21], whereas FMDV serotype A, lineage Euro-SA, was isolated from infected cattle on another farm during routine surveillance in 2022 [22]. Shahein et al. evaluated the circulating virus during an outbreak in 2022 in samples collected from infected animals [23]. The obtained sequences belonged to FMDV serotype A (African topotype), which originated from the ancestor prototype Sudan/77. The divergence from local isolates from 2020 was 9.3%. Also, serotype A-African topotype-G-IV was isolated from dairy farms in Behera and Gharbiya during the period between July and October 2022 despite the animals being vaccinated with a local polyvalent inactivated vaccine developed by the Veterinary Serum and Vaccine Research Institute (VSVRI) in Cairo, Egypt [24].viruses-16-01536-t001_Table 1Table 1FMDV serotypes in Egypt.YearSerotype/TopotypeReference1950SAT2[16]1964–2005O[14]1972A[14]2006A/Africa/G-VII[14]2010/2011A/Asia/Iran05O/ME-SA/PanAsia-2O/EA-3[3]2012SAT2/VII/Ghb-12SAT2/VII/Alx-12A/Africa/G-IV[3,16]2016O/ME-SA/Sharqia-72O/EA-3A/Africa/G IV[18]2019SAT2/VII, Lib12[18]2020,2022A/Africa/G IVO/Euro-SA topotypeA, lineage EURO-SA[23][21,22]2022A-African topotype-G-IV[24]

### 2.6. Challenges in Control

FMD control varies on the basis of disease status, and FMD-free countries focus on decreasing the risk of virus incursions from adjacent and trade-partner nations by regulating animal and animal product movements. In endemic countries, FMD control is achieved via accurate and rapid diagnosis, regular checks, and compulsory mass vaccination with constant surveillance to evaluate the FMDV serotypes circulating with the vaccine strain [25].

In Egypt, the control of FMD relies mainly on quarantine and mass vaccination. Mandatory vaccination of all ruminants every six months and of dairy cows every four months is achieved via a trivalent vaccine, which is prepared from local isolates and comprises strains of serotypes A, O, and SAT2 [26]. Despite the systematic use of vaccination, severe annual outbreaks of FMD have been documented in different localities in Egypt, and there is evidence of continued FMDV circulation in vaccinated animal populations [27].

According to El Ashmawy et al., medium-sized beef producers in Egypt are at high risk of FMD outbreaks because of the high turnover and continuous influx of replacements from live animal markets, where biosecurity practices are limited for budgetary reasons [28]. On the other hand, as a developing country, Egypt depends on the importation of meat and live animals to narrow the gap between production and consumption. Egyptian trade relationships have increased during the last decade, especially in terms of livestock imports. The trade routes included imports from Sudan, India, and South America, specifically Brazil and Colombia, according to a report issued by the USDA in 2021 [22]. This upsurges the chance of introducing new FMDV strains through the movement of FMD-infected animals across international borders. Both Asian and African FMDV strains have been introduced into Egypt, which is facilitated by their unique transcontinental location [19]. In accordance with Shahein et al., the vaccine regime in the country was modified to include serotype O pan-Asian II (EGY/2010), A Iran 05 (A/EGY/1/2012), SAT2 (EGY/Gharbia/2012), and SAT2 (LIB/2018), along with the isolated serotype A G-IV of 2020 (Acc. No. MW413350) [23].

## 3. Lumpy Skin Disease

### 3.1. General Information

Lumpy skin disease (LSD) is a devastating infectious disease of cattle of all ages and breeds [29]. Infection with LSDV ranges from inapparent to severe disease in cattle [30,31]. Blood-sucking arthropods, mosquitoes, and ticks transmit the virus from one animal to another [32]. LSD is a generalized and epitheliotropic disease causing vasculitis, lymphadenitis, and inflammatory nodules on the skin [33]. Clinical signs include fever; multiple circumscribed raised nodules in the skin; necrotic plaques in the mucous membranes of the mouth, respiratory tract, vulva, and prepuce; lymphadenitis; and edema of the limbs and ventral parts of the body [33,34,35] (Figure 4). The three main features of LSD skin lesions are the depth of the nodule, the development of the sit-fast (inverted conical necrosis), and the random distribution. All cattle breeds and age groups can be affected. However, Bos taurus is more prone to developing clinical signs compared to Bos indicus. Young animals and high-producing dairy breeds with fine skin are most susceptible to LSDV [34].

### 3.2. Economic Impact

The disease causes significant economic loss due to hide damage, decreased weight gain, loss of milk production, mastitis, infertility in lactating cows, infertility in bulls, and death [30] (Figure 5). In cattle that recover after long-term illness, there are long-term signs of mastitis, pneumonia, and deep holes in the hide [36]. Owing to severe production losses in cattle and its rapid spread, LSD is considered a “List A” disease by the OIE. The morbidity rate of the disease may range from 3% to 85% depending on the immune status of the host and the abundance of mechanical vectors [31]. In endemic areas, the morbidity rate is usually around 10%, while the mortality rate rarely exceeds 5% [37]. Indirect losses include bans on the international trade of livestock, the costs associated with mitigation and control efforts, and reduced consumer confidence.

### 3.3. The Virus

Lumpy skin disease virus (LSDV) belongs to the *Capripox* genus within the family *Poxviridae*, subfamily *Chordopoxvirinae* (poxviruses of vertebrates), which shares a group-specific antigen [nucleoprotein (NP) antigen] [38]. Other closely related viruses of the same genus are sheep pox virus (SPPV) and goat pox virus (GTPV) [39]. The LSDV genome consists of a central coding region surrounded by identical inverted terminal repeats (2.4 kbp) and encloses 156 putative genes [40]. Among them are 146 conserved genes that encode proteins involved in several tasks (DNA replication, transcription and mRNA biogenesis, protein processing, virion structure and assembly, nucleotide metabolism, and viral virulence and host range) [41]. The specific genes within CaPV that are responsible for genetic differences are known as G-protein-coupled chemokine receptor (GPCR) genes [42]. Initially, GPCR genes were considered stable, but they have since become a cause for concern due to the high frequency of reported synonymous mutations (caused by natural drift) and non-synonymous mutations associated with highly cell-passaged viruses [43]. A total of nine genes in LSDV, which specifically adapted for cattle infection and are inactive in SPPV and GTPV, were reported [38]. The persistent nature of LSDV was fully studied, where the virus appears to be resistant to a wide range of physical and chemical components and can persist viably in desiccated skin crusts (35 days), skin necrotic nodules (>33 days), and in air-dried hides (18 days) [44]. In the environment, LSDV can persist for several months, particularly in dark conditions in contaminated animal sheds [3].

### 3.4. Global Burden

Geographically, LSD was first described in Zambia in 1929 [45]. Between 1943 and 1945, the disease spread to other southern African countries [46]. In 1957, LSD was first observed in East Africa in Kenya. In the period 1970–1990, it occurred in most Central and West African countries [47].

According to Hunter and Wallace, LSD is widespread and endemic throughout Africa, excluding Libya, Tunisia, Algeria, and Morocco [48]. However, Libya reported its first cases of LSD on cattle farms in the northwestern part of the country in 2023, and the last report on LSD in North Africa in July 2024 detailed the detection of LSD in Algeria for the first time. LSD in Tunisia was first reported by the World Organization for Animal Health (WOAH) on 14 August 2024. Currently, Morocco is the only country where LSD cases have not been reported.

Outside of Africa, LSD was reported for the first time in the Middle East (ME), where unconfirmed cases of the disease were reported in Oman and Kuwait between 1984 and 1986 [49]. Later, the disease was identified in Israel (1989 and 2006) [50]; Saudi Arabia (1992) [51]; Oman (2009) [52]; Lebanon, Jordan, and Turkey (2012and2013) [53]; and Iran and Iraq (2014) [54]. This was followed by outbreaks in Azerbaijan (2014), Armenia (2015), and Kazakhstan (2015), the southern Russian Federation (Dagestan, Chechnya, Krasnodar Kray, and Kalmykia), and Georgia (2016) [29]. Since 2014, LSD has advanced into the northern part of Cyprus, Greece (2015), Bulgaria, the former Yugoslav Republic of Macedonia, Serbia, Montenegro, Albania, and Kosovo (2016) [29]. In 2018, no outbreaks of LSD were reported in the Balkan region, after a decline in the number of outbreaks reported in 2017 (385) compared with 2016 (7483) [55]. In 2018, LSD outbreaks were reported only in Russia (63 outbreaks), Turkey (46 outbreaks), and Georgia (6 outbreaks) between April and November [55]. LSD was reported in Yi li, Xinjiang Province, China, in 2019. In the following 2 years, the disease spread to southern and eastern parts of China and countries in South Asia, including Nepal, Bhutan, Vietnam, Thailand, and Myanmar [41,56]. By 2022, the disease had spread east and north to Mongolia and eastern Siberia [40].

### 3.5. Egyptian Situation

There is no doubt that the movement of infected live animals plays an important role in the spread of diseases over long distances, which is what has happened in Egypt. LSD was reported for the first time as an exotic disease in Egypt in May 1988 after the importation of apparent healthy cattle from Somalia [49]. On 2 June 1988, 14 out of 194 Holstein cattle at a government farm near the quarantine station in Suez, Egypt, were affected by the disease [57]. On 30 October 1988, a total of 29 native cows, 7 fattening steers, and 3 heifers showed skin eruptions similar to those that appeared in Suez; these eruptions appeared in the villages of Tel-El-Baar and Tel-Abohomed of El-Tel-El-Kabir in Ismailia Governorate, after which the disease spread further to three neighboring villages called El-Zaheria, Ezbet El-Arab, and Kafer Ghonaim [57]. Between 10th April and 23rd August 1989, LSD was recorded in all Egyptian governorates except Qena, Sohag, Red Sea, Matroh and South Sinai Governorates and had a severe effect, as 1449 animals died [44]. The rapid response to contain the outbreak was directed toward the vaccination of nearly two million cattle with a sheep pox vaccine [30].

Sporadic cases of LSD were subsequently reported in cows in Egypt in 1990 (El-Menia, El-Fayoum, Aswan, Sohag, Beni Suef, and Qena Governorates), 1992 (Nag-Hamadi, New Valley, and Assiut), and 1998 (nine villages in El-Menia Governorate) [58,59].

LSD re-emerged in Egypt in 2005 and 2006 and was introduced into Egypt by infected cattle imported from Ethiopia [60]. Despite an extensive vaccination campaign, the disease rapidly spread throughout the country. It has been reported in Damietta Governorate, Gharbia Governorate, El-Kaluobia, Alexandria, Beni Suef, Al-Fayium, and Giza Governorate [44]. The disease reappeared again in 2011 and 2014 [61,62,63]. Later, sporadic outbreaks of LSD were subsequently reported in Egypt. A retrospective study by Ezzeldin et al. (2023) revealed that the total number of positive LSD-related outbreaks reached 577 between 2006 and 2018 [64]. From 2019 to January 2020, Ali et al. (2021) reported LSD in three governorates (Sharkia, Dakahlia, and Kaloubia) in the Nile Delta, Egypt [65], whereas Abed EL Naby et al. (2021) reported LSDV infections in cattle in the El-Wady El-Gedid Governorate from August 2020 to early 2021 [66]. In April and May 2023, El-Hamady et al. detected LSDV in five skin nodule samples collected from cattle suspected to have LSD in three governorates in Egypt (Dakahlia, El-Menia, and El-Fayoum) using Taq-Man real-time PCR [67]. Figure 6 shows the LSDV sequences registered in GenBank from Egypt between 1989 and 2022. Table 2 shows LSD outbreaks in different localities in Egypt.

### 3.6. Challenges in Control

Vectors and the movement of infected cattle are the main routes by which LSD spreads into uninfected areas. Hence, vaccinating and restricting livestock movement are key control actions [30]. Despite regular LSD vaccination of cattle, vaccination failures and recurrences of multiple outbreaks have been recorded in Egypt [68]. Incomplete vaccination coverage, inappropriate conditions for vaccine storage and transport, and the presence of pre-existing immunosuppressive diseases could also be among the possible reasons for LSDV outbreaks [33]. Partial protection of cattle against LSD has been achieved in Egypt via an attenuated Romanian strain of the SPV vaccine [69]. This necessitates the use of an alternative strategy in the use of the homologous LSDV vaccine. A strategic plan is strongly recommended, particularly one that warrants mass coverage of the vaccination among cattle populations all over Egypt. Bazid et al. evaluated the safety and efficacy of a new live-attenuated LSD vaccine based on the Neethling strain vaccine produced by Middle East for Vaccines (MEVAC^®^, Cairo, Egypt). The acquired results reveal that the MEVAC^®^ LSD vaccine produced minor clinical signs in the field (around 0.6% of the animals exhibiting minor skin inflammatory reactions, which disappeared within a few hours), and vaccinated calves were resistant to challenge [70].viruses-16-01536-t002_Table 2Table 2LSD outbreaks in different localities in Egypt.YearLocalityReferencesJune 1988October 1988Farm near the quarantine station in Suez, the village of Tel El Kabir, Ismalia[57]May 1988 and May 1989Ismailia and Sharkia Governorates[49]1988El-Menia Governorate[59]1989Dakahlia[71]1992Nag-Hamadi, New Valley, and Assiut[58]2004–2007Lower Egypt[72]2006EL Kanater-Kaluobia and El-Noubaria-Alexandria, Giza Governorate, and Damietta province[44,73,74]2008Dakahlia Governorate[75]2011El Menoufia and El Sadat city[76]2012Behera and Alexandria Governorates[61]2014Beni Suef[62]January 2014 to mid-2015Sharqia[77]May 2015 and August 2016Beni Suef Governorate[78]June 2015 to September 2016Ismailia and Beni-Suef[33]2017Beni Suef, El-Fayoum El Giza, El-Menia, El-Gharbia, El-Qalyubia, and Sharkia[43]2017Beni Suef[32]2017–2018Beni Suef, Sohag and Aswan Governorates[33]2019Sohag[79]2019Elwasta, Beni Suef[68]2020El-Wady El-Gedid Governorate[66]2020–2021Alexandria, Beheira, Gharbia, Kafr El-Sheikh, Menofia, and Qalyubia[36]2022Beni Suef[69]2023Giza, Cairo, Fayoum, Beni Suef, Menia, Sohag, Assuit, Qena, Louxer, Aswan, Delta (Qaliobya, Monofya, Sharkya, Gharbya, Dakahlia, Domyat, Kaferelshikh, and Alexandria), Canal Suez, Portsaid, Ismailia, Red Sea Desert, New Valley, and Matrouh[64]2023Dakahlia, El-Menia, and El-Fayoum[67]

## 4. Bovine Ephemeral Fever (Three-Day Stiff Sickness)

### 4.1. General Information

Bovine ephemeral fever (BEF) is an arthropod-borne disease of cattle and water buffaloes [80,81]. The disease is also known as 3-day sickness, bovine enzootic fever, bovine influenza, or stiffseitke. The characteristic clinical signs of BEF are a sudden onset of bi-phasic fever, sudden and severe drop in milk production, anorexia, depression, ocular and nasal discharges, salivation, muscle stiffness, lameness, ruminal stasis, sternal recumbency, and other inflammatory responses [82]. Pulmonary and subcutaneous emphysema with the accumulation of air under the skin over the backbone, shoulders, or neck may develop, which is referred to as atypical three-day stiff sickness. (Figure 7). Sometimes pharyngeal paralysis occurs due to the hyaline degeneration of muscles.

### 4.2. Economic Impact

BEF is considered a disease that has economic impacts, such as loss of milk production in both quality and quantity, a loss of body condition in beef herds, infertility, and abortion [83] (Figure 8). Some animals may become recumbent for 24 h, while others may develop persistent recumbency due to permanent nerve damage and degenerative changes in the spinal cord [82]. BEF is characterized by high morbidity rates of up to 100%; however, the mortality rate is generally low (rarely exceeds 1%), but cattle in good condition are usually affected more severely, and the mortality rate can reach 30% in very fat cattle [80].

### 4.3. The Virus

Bovine ephemeral fever virus (BEFV), a negative-sense, single-stranded RNA, is classified as a member of the genus *Ephemerovirus* in the *Rhabdoviridae* family in the order *Mononegavirales* [84]. BEFV has the typical rhabdovirus bullet-shaped morphology. The BEF genome contains ten long open reading frames arranged in the order 3′-N-P-M-G-[GNS-α1-α2-β-γ]-L-5′. The virion surface glycoprotein (G) is responsible for cell attachment and entry; it is also a type-specific neutralizing antigen and induces protective immunity in cattle [85].

Phylogenetically, BEFV is classified into three lineages—East Asian, Australian, and Middle Eastern—based on the analysis of the glycoprotein G gene [86]. According to antigenic studies, BEFV exists as a single serotype, and BEF viruses located in different lineages are antigenically related [85].

Numerous closely related ephemeroviruses (including Berrimah virus, Kotonkan virus, Kimberley virus, Malakal virus, Mavingoni virus, Adelaide River virus, Puchong virus, Koolpinyah virus, Hefer Valley virus, and Obodhiang virus) have been recognized. However, among these, only the Kotonkan virus isolated in Nigeria and the Hefer Valley virus isolated in Israel have been associated with clinical ephemeral fever in cattle.

### 4.4. Global Burden

The disease was found to be most prevalent in humid and plain areas. BEFV infection has been reported in tropical, subtropical, and temperate regions of Africa, Asia, and Australia [86,87]. It reported to be occurring from southern Africa to the Nile River Delta, in the Middle East and Asia particularly, in South and Southeast regions, and in northern and eastern Australia [82]. China, Taiwan, the Korean Peninsula, and southern Japan also reported the existence of the disease. BEFV does not occur in the islands of the Pacific, Europe (except Turkey), and the Americas [85].

### 4.5. Egyptian Situation

Bovine ephemeral fever is one of the most important infectious diseases of cattle and water buffaloes in Egypt. Egypt is one of the countries in which BEF was first described. The disease has persisted in Egypt for more than a century (Table 3). BEF was first detected in Egypt by Piot in 1895 [88]. It is known as dengue fever in cattle, but the first detailed description of the disease was an outbreak in 1909 that originated in Aswan, where it traveled down the Nile Valley to Cairo and spread across the Delta to the coast [80]. Successive outbreaks affecting hundreds of cattle were documented in 1915 and 1919–1920, in addition to outbreaks in 1990–1991, 2000–2001, and 2004–2005 [89]. A major outbreak from 1990 to 1991 affected cattle throughout the country, moving along the Nile Valley from Upper Egypt (in the summer of 1990) to the eastern part of the Delta (in the autumn) [89]. In 1991, the disease affected imported cattle (*n* = 250,000) and a smaller number of native cattle and water buffaloes all along the Nile Valley, in the Delta, and at several oases west of the Nile. Morbidity rates vary from 20 to 90%, and mortality rates in imported animals range from 1.5 to 3.0%. The source of the virus was thought to have been the airborne displacement of vectors from areas in southern or eastern China [89]. Lately, the disease was recorded during the summers of 2010, 2014, 2017, 2018, and 2019 [90,91,92]. EL-Allawy et al. (2021) examined a total of 156 cattle and buffaloes (72 Frezian, 33 native breed cattle, and 51 buffaloes) of different ages (6 months to 4 years) and sexes from January 2018 to September 2019 [58]. These animals were from different localities in the Assiut, Sohag, and El-Menia Governorates. The investigated animals were characterized by the sudden onset of fever in 25.64% (40/156) of the animals. Bovine ephemeral fever is endemic in Egypt, with repeated periodic outbreaks. Most of the affected governorates are around the Nile Delta. Figure 9 shows the BEFV sequences registered in GenBank from Egypt between 2000 and 2017.

### 4.6. Challenges in Control

Multiple episodes of clinical ephemeral fever in the same cattle have been reported in the field [90]; however, it is not recognized if other ephemeroviruses might have been responsible for the disease manifestation. A strong neutralizing antibody response after natural BEFV infection or vaccination was reported; however, Della-Porta and Snowdon (1979) reported no correlation between the level of the neutralizing antibody response and protection and suggested that the cell-mediated responses may also be requisite [93]. Under such conditions, attenuated vaccines are endorsed, but it was reported that they may cause the transmission of infections from vaccinated cattle to susceptible ones by insects. Concerning inactivated vaccines, it was observed that a loss of virulence is associated with a loss of immunogenicity. Inaba et al. (1974) found that consecutive vaccinations with a live-attenuated virus followed by an inactivated (killed) virus resulted in a stronger and more durable neutralizing antibody response than vaccination with a live-attenuated vaccine alone or with two doses of the inactivated vaccine [94]. The locally produced live-attenuated BEFV vaccine, which is inactivated just before inoculation via reconstitution in PBS containing saponin, is used for the prevention and control of BEF in Egypt [95].

## 5. Conclusions

In this review, the current statuses of FMD, LSD, and BEF in Egypt were summarized. Three FMDV serotypes (A, O, and SAT2) have been circulating in Egypt. The movement of FMD-infected animals across international borders increases the risk of introducing new FMDV strains into the country. Asian and African FMDV strains have been introduced into Egypt and, ultimately, the Euro-SA topotype. LSD is a devastating infectious disease affecting cattle of all ages and breeds. Incomplete vaccination coverage, improper vaccine storage and transport, and pre-existing immunosuppressive diseases may contribute to LSDV outbreaks. Egypt is one of the countries in which BEF was first described. The disease has persisted in Egypt for more than a century. Only one serotype of LSDV and BEFV has been identified so far. These three diseases are considered endemic in Egypt, leading to severe economic losses.

## Figures and Tables

**Figure 1 viruses-16-01536-f001:**
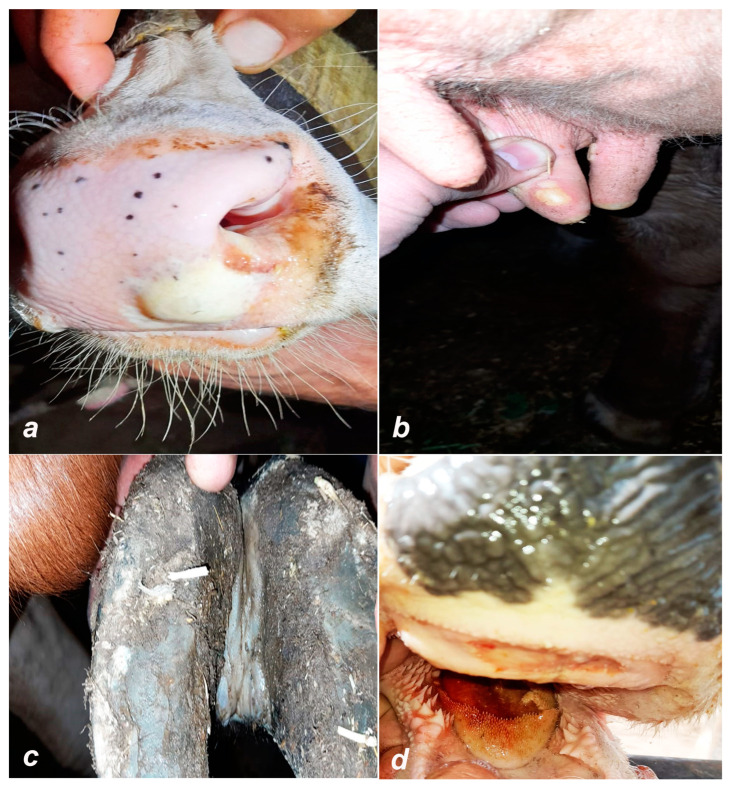
Clinical signs of foot and mouth disease: (**a**) vesicle on the muzzle, (**b**) vesicle on the teat, (**c**) vesicle in interdigital space, and (**d**) ruptured vesicle on the tongue.

**Figure 2 viruses-16-01536-f002:**
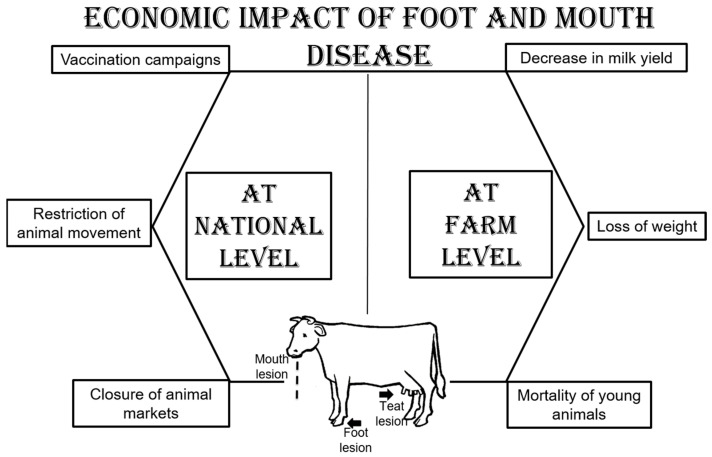
Economic impact of foot and mouth disease.

**Figure 3 viruses-16-01536-f003:**
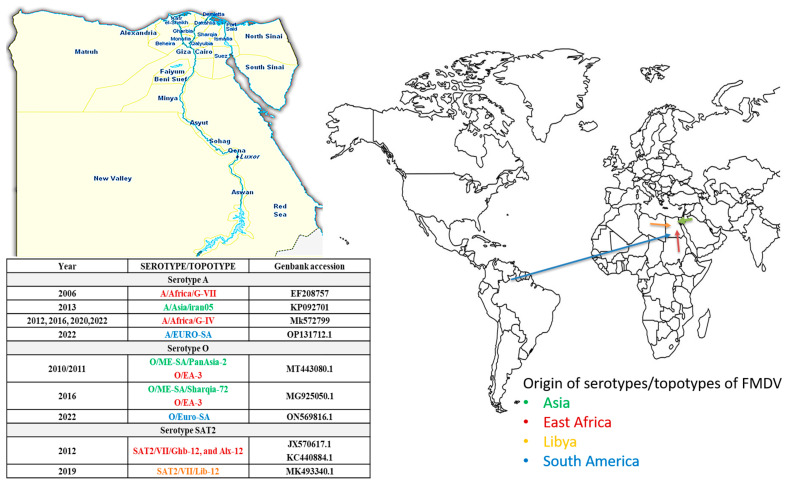
Circulating serotype/topotype of FMDV in Egypt.

**Figure 4 viruses-16-01536-f004:**
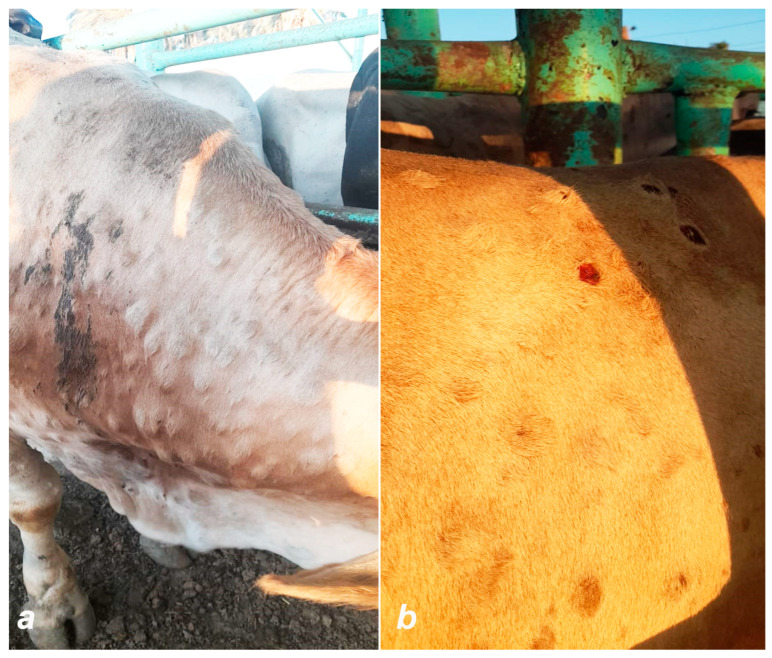
Clinical signs of LSD: (**a**) circumscribed raised nodules in the skin and (**b**) sit-fast lesions.

**Figure 5 viruses-16-01536-f005:**
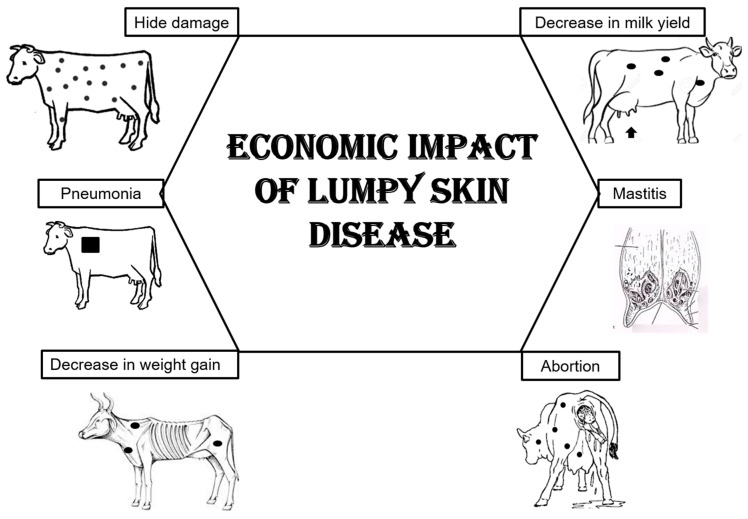
Economic impact of lumpy skin disease.

**Figure 6 viruses-16-01536-f006:**
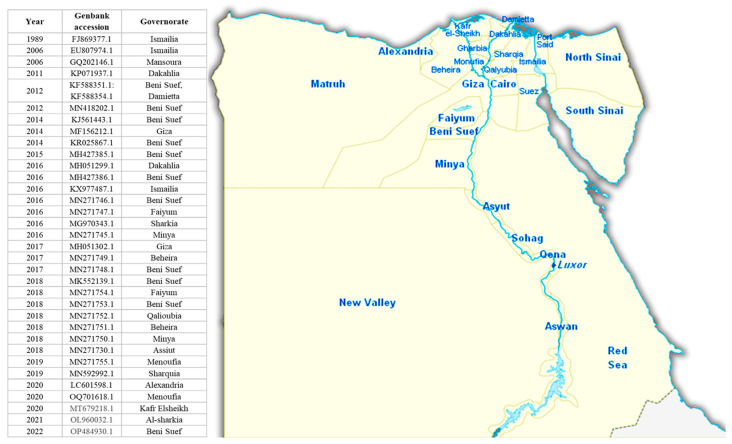
The LSDV sequences registered in GenBank from Egypt between 1989 and 2022.

**Figure 7 viruses-16-01536-f007:**
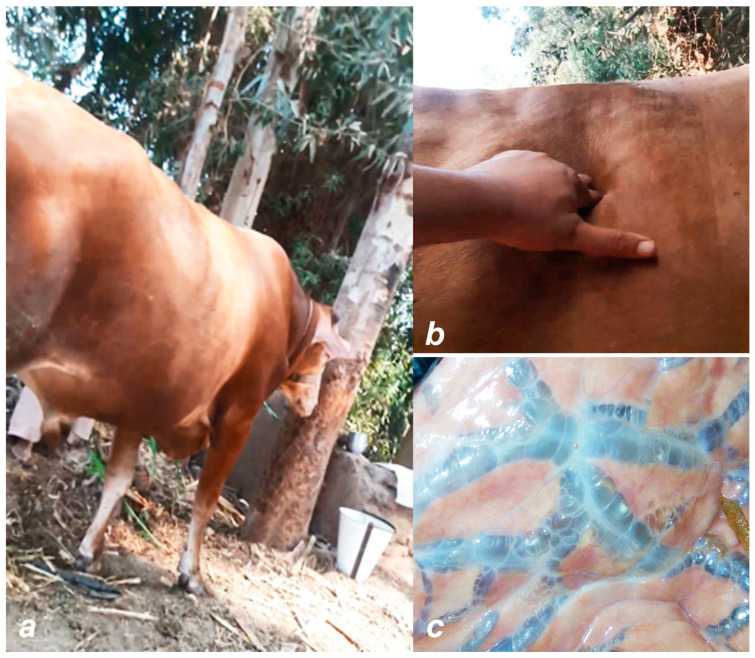
Atypical three-day stiff sickness: subcutaneous (**a**,**b**) and pulmonary emphysema (**c**).

**Figure 8 viruses-16-01536-f008:**
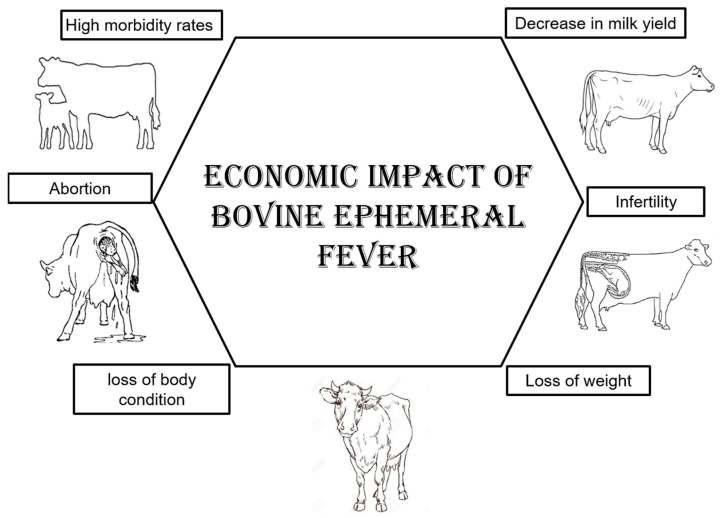
Economic impact of bovine ephemeral fever.

**Figure 9 viruses-16-01536-f009:**
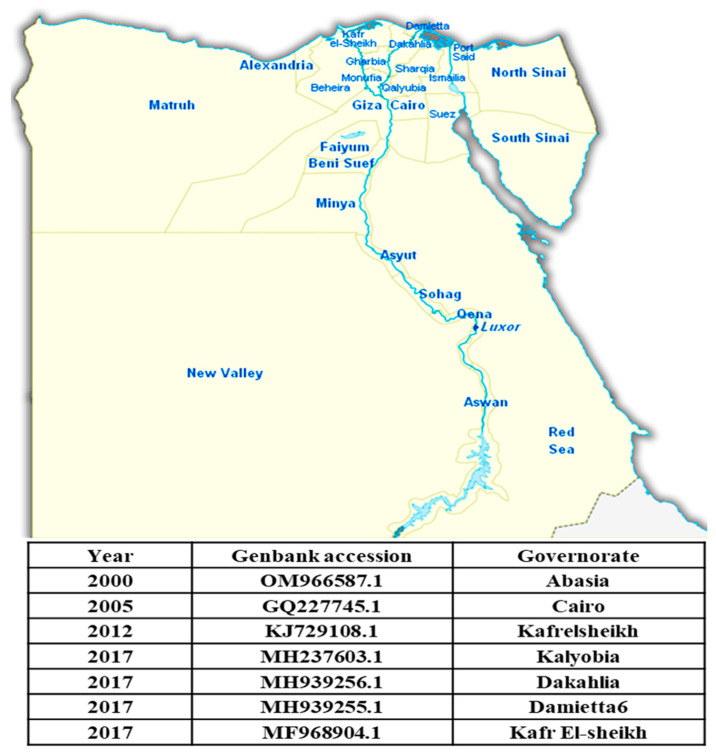
The BEFV sequences registered in GenBank from Egypt between 2000 and 2017.

**Table 3 viruses-16-01536-t003:** BEF history in Egypt.

BEF in Egypt	
1895 as dengue fever	[88]
1909 at Aswan	[88]
1915 and 1919–1920	[89]
1990–1991/2000–2001 and 2004–2005	[89]
2010	[90]
2014	[91]
2017, 2018, and 2019	[58,92]

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
