# Peer review of "The Current Epizootiological Situation of Three Major Viral Infections Affecting Cattle in Egypt"

_viruses, 2024, doi:10.3390/v16101536_

Round 1

Reviewer 1 Report

Comments and Suggestions for Authors

Recently, I reviewed the article entitled " The Current Epizootiological Situation of Three Major Viral Infections Affecting Farm Animals in Egypt" written by Rouby et al.  I expected to see more specific information on the topic. Unfortunately, the article only provided summarized data on the three diseases in the tables, without additional details on the viruses that allows to understand the epidemiological situation in the country, as phylogenetic analyses of the presented viruses/strains, and maps. I anticipated finding maps showing registered outbreaks for each disease, at least for the current time (the past few years or the last decade). I was puzzled, why the authors presented data on FMD in Egypt up to 2022, when data on BEFV -only up to 2019. Whether more recent data is available or registered, or not? This data should be also included into the text (2020-2024)

 Title and keywords

Neither the title, nor keywords provide enough information allowing to search the current article/data on the theme

Abstract.

In general, the abstract does not correspond with the presented data. The abstract is not informative: as it lacks background, study details, and conclusions. Only the last sentence provides some minimal information.

Introduction

I strongly suggest to subdivide the introduction about the specific diseases into subsections as, for example “general information/ data on the disease in the world, clinical signs, control or/and vaccination, history, current situation” etc.  All the study based on registered outbreaks or cases, isn’t it?  The source (the registered outbreak based on laboratory diagnosis (which one: serological or molecular, both of them)/clinical sings) should be straightly defined.

Conclusion sections

A conclusion section is absent at all.

Reference

Citation do not meet the requirements of the journal both in the text and in the reference section.

Line 75. Extra dot.

Line 77. “whereas serotypes (SAT1, 2 and 3) currently circulate in Africa”. The information is incomplete. Now it was registered in the Middle East region also.

Lines 120-124. I suggest to transfer the data into the beginning of introduction about FMD.

Lines 216-217. “imported into Egypt with foot and mouth disease by infected cattle imported from Ethiopia”- I doubled whether both diseases were imported simultaneously. According to the presented by the authors reference (El-Kholy et al., 2008), there were spoken on the LSDV only. The reference is not suitable to the context.

Line 257-258. “genus  Ephemerovirus in the Rhabdoviridae family in the order Mononegavirales”. Mononegavirales, Ephemerovirus,  Rhabdoviridae- should be written in Italic

Line 261. “BEFV infection occurs seasonally in tropical,”- I disagree with the statement. It causes outbreaks not anually. The statement should be changed

Lines 265-266. “BEF was first detected in Egypt by Piot in 1895.” The reference is absent.

Line 284-285.  Bovine ephemeral fever” should be changed to “BEF”

Lines 263-286. Connections and sequence of events and their descriptions are not well written. The changes have to be done. No data on prevention and control is written at all both in the world and in Egypt.

Line 354. “arecent” to change with “ a recent”.

Author Response

Reviewer #1

Comment

Recently, I reviewed the article entitled " The Current Epizootiological Situation of Three Major Viral Infections Affecting Farm Animals in Egypt" written by Rouby et al. I expected to see more specific information on the topic. Unfortunately, the article only provided summarized data on the three diseases in the tables, without additional details on the viruses that allows to understand the epidemiological situation in the country, as phylogenetic analyses of the presented viruses/strains, and maps. I anticipated finding maps showing registered outbreaks for each disease, at least for the current time (the past few years or the last decade). I was puzzled, why the authors presented data on FMD in Egypt up to 2022, when data on BEFV -only up to 2019. Whether more recent data is available or registered, or not? This data should be also included into the text (2020-2024)

Response: thank you for your kind comment. It was changed as recommended. Three maps were added, and information was updated. Unfortunately, no data about BEF after 2019 are available, and in Genbank, the last submitted sequences are from 2017. Data on LSD was updated till the recently published paper from Egypt in 2023.

Comment

Title and keywords

Neither the title, nor keywords provide enough information allowing to search the current article/data on the theme

Response: thank you for your kind comment. It was changed as recommended.

Abstract.

In general, the abstract does not correspond with the presented data. The abstract is not informative: as it lacks background, study details, and conclusions. Only the last sentence provides some minimal information.

Response: thank you for your kind comment. It was changed as recommended

Introduction

I strongly suggest to subdivide the introduction about the specific diseases into subsections as, for example “general information/ data on the disease in the world, clinical signs, control or/and vaccination, history, current situation” etc.  All the study based on registered outbreaks or cases, isn’t it?  The source (the registered outbreak based on laboratory diagnosis (which one: serological or molecular, both of them)/clinical sings) should be straightly defined.

Response: thank you for your kind comment. It was changed as recommended

Conclusion sections

A conclusion section is absent at all.

Response: thank you for your kind comment. It was added as recommended

Reference

Citation do not meet the requirements of the journal both in the text and in the reference section.

 Response: thank you for your kind comment. It was edited as recommended

Line 75. Extra dot.

Line 77. “whereas serotypes (SAT1, 2 and 3) currently circulate in Africa”. The information is incomplete. Now it was registered in the Middle East region also.

Response: thank you for your kind comment. Information was added as recommended

Lines 120-124. I suggest to transfer the data into the beginning of introduction about FMD.

Lines 216-217. “imported into Egypt with foot and mouth disease by infected cattle imported from Ethiopia”- I doubled whether both diseases were imported simultaneously. According to the presented by the authors reference (El-Kholy et al., 2008), there were spoken on the LSDV only. The reference is not suitable to the context.

Response: thank you for your kind comment. It was changed as recommended

Line 257-258. “genus  Ephemerovirus in the Rhabdoviridae family in the order Mononegavirales”. Mononegavirales, Ephemerovirus,  Rhabdoviridae- should be written in Italic

Line 261. “BEFV infection occurs seasonally in tropical,”- I disagree with the statement. It causes outbreaks not anually. The statement should be changed

Response: thank you for your kind comment. It was changed as recommended

Lines 265-266. “BEF was first detected in Egypt by Piot in 1895.” The reference is absent.

Response: thank you for your kind comment. It was added as recommended

Line 284-285.  “Bovine ephemeral fever” should be changed to “BEF”

Lines 263-286. Connections and sequence of events and their descriptions are not well written. The changes have to be done. No data on prevention and control is written at all both in the world and in Egypt.

Line 354. “arecent” to change with “ a recent”.

Response: thank you for your kind comment. It was changed as recommended.

Reviewer 2 Report

Comments and Suggestions for Authors

What I find missing from this work is a conclusion in the form of interactions between the pathogens described. It would also have been necessary to mention the management of infected animals and the ways in which official veterinarians control these entities. 

Author Response

Comment:

What I find missing from this work is a conclusion in the form of interactions between the pathogens described. It would also have been necessary to mention the management of infected animals and the ways in which official veterinarians control these entities.

Response: thank you for your kind comment. It was added as recommended.

Reviewer 3 Report

Comments and Suggestions for Authors

 Dear authors,

I found just one point I did not like in the manuscript.  

When talking about FMDV a lot of attention was paid to serotypes, and what about serotypes of LSD and BEFV? It would be good that all three viruses would be described according to the same plan or structure.

Minor remarks:

Fig. 1 – what do arows mean at the bottom of the cow silhouette?

Line 64

The virus exists in seven serologically and genetically divergent serotypes: O, A, C, southern African territories (SAT1-3),

maybe it should be

… : O, A, C, in southern African territories (SAT1-3),…

Line 167

Capripox should be italic

Don`t You think that Fig. 3 should be stylistically similar to Fig 2 and Fig. 1.?

Lines 256-257 Latin names should be in italic.

Author Response

Dear authors,

I found just one point I did not like in the manuscript.  

When talking about FMDV a lot of attention was paid to serotypes, and what about serotypes of LSD and BEFV? It would be good that all three viruses would be described according to the same plan or structure.

Response: thank you for your kind comment. Only one serotype of BEFV and LSDV has been identified so far. This information was added to the paper.

Minor remarks:

Comment: Fig. 1 – what do arows mean at the bottom of the cow silhouette?

Response: signs of FMD; Vesicles (fluid-filled blisters) on the feet, particularly between the toes, in the mouth, especially on the lips, tongue, and in mammary glands. A plate with field cases illustrates the clinical picture of FMD was added.

Comment:

Line 64

The virus exists in seven serologically and genetically divergent serotypes: O, A, C, southern African territories (SAT1-3), maybe it should be … : O, A, C, in southern African territories (SAT1-3),…

Response: O, A, C, southern African territories (SAT1-3), and Asia-1.

Comment:

Line 167

Capripox should be italic

Response: thank you for your kind comment. It was changed as recommended

Comment:

Don`t You think that Fig. 3 should be stylistically similar to Fig 2 and Fig. 1.?

Lines 256-257 Latin names should be in italic.

Response: thank you for your kind comment. It was changed as recommended.